# Effects of Weaning Age at 21 and 28 Days on Growth Performance, Intestinal Morphology and Redox Status in Piglets

**DOI:** 10.3390/ani11082169

**Published:** 2021-07-22

**Authors:** Dongxu Ming, Wenhui Wang, Caiyun Huang, Zijie Wang, Chenyu Shi, Jian Ding, Hu Liu, Fenglai Wang

**Affiliations:** 1State Key Laboratory of Animal Nutrition, College of Animal Science and Technology, China Agricultural University, Beijing 100193, China; mdx920825@163.com (D.M.); wangwh1025@cau.edu.cn (W.W.); huangcaiyun@cau.edu.cn (C.H.); wzjgf191114@163.com (Z.W.); scyshichenyu@163.com (C.S.); wangfl@cau.edu.cn (F.W.); 2National Animal Husbandry Service, Building No. 20, Maizidian Street, Beijing 100125, China; Dingjian1029@126.com

**Keywords:** weaned pigs, weaning age, growth performance, redox status, intestinal morphology

## Abstract

**Simple Summary:**

After weaning, pigs are subjected to a variety of nutritional, psychological and environmental stresses. Historically, weaning age was determined knowing that antibiotics could be included in postweaning diets for piglets. The use of antibiotic growth promoters to help prevent weaning stress in weaned pigs has been forbidden in Japan, Korea, the European Union and China. In this study, intestinal morphology, pH of the stomach and antioxidant status of pigs weaned at 28 d were better than pigs weaned at 21 d. These results indicated that strategies including enhancing the intestinal absorption function and antioxidant ability in weaned pigs can improve growth performance and decrease diarrhea incidence.

**Abstract:**

The study objective was to assess effects of different weaning ages on growth performance, intestinal morphology and redox status in Duroc × Landrace × Large White piglets (n = 96) fed diets without antibiotic growth promoters. Piglets were selected from 24 litters based on similar body weight at 14 d of age. All piglets were allocated to two groups in a completely random design with six replicates and eight pigs per replicate (four barrows and four gilts), which were weaned at 21 (n = 48; BW = 6.87 ± 0.33 kg) and 28 (n = 48; BW = 8.49 ± 0.41 kg) days of age. After weaning, pigs were fed a corn–soybean meal-based diet. Average daily gain (ADG), average daily feed intake (ADFI), feed conversion (F:G), diarrhea incidence, gastrointestinal pH, intestinal morphology and redox status were determined. Pigs weaned at 28 d displayed increased ADG from d 8 to 14 (*p* < 0.01) compared with pigs weaned at 21 d. Pigs weaned at 28 d had a higher ADFI from d 0 to 7 (*p* < 0.01), d 8 to 14 (*p* < 0.01), d 15 to 28 (*p* < 0.05) and during the entire experimental period (*p* < 0.01) compared with pigs weaned at 21 d. Pigs weaned at 21 d had an improved F:G from d 15 to 28 (*p* < 0.05) compared with pigs weaned at 28 d. Pigs weaned at 28 d had decreased diarrhea incidence from d 8 to 14 (*p* < 0.01) and the entire experimental period (*p* < 0.01) compared with pigs weaned at 21 d. On d 28, the pH of the stomach contents in pigs weaned at 21 d was significantly higher compared with pigs weaned at 28 d (*p* < 0.01). On d 14, the morphology of the duodenum, jejunum and ileum in pigs weaned at 28 d was improved compared with pigs weaned at 21 d. During the experiment period, the antioxidant abilities of pigs weaned at 28 d of the heart, liver, kidney, intestinal and serum were better than pigs weaned at 21 d. In conclusion, intestinal morphology, pH of the stomach and antioxidant status of pigs weaned at 28 d were better than pigs weaned at 21 d. These factors supported better growth performance and decreased diarrhea incidence.

## 1. Introduction

Weaning is a necessary step in pig production. After weaning, pigs are subjected to a variety of nutritional, psychological and environmental stresses [1,2]. Historically, weaning age was determined knowing that antibiotics could be included in postweaning diets for piglets. Widespread use of antibiotic growth promoters has facilitated the development of antibiotic-resistant bacteria and increased chances of antibiotic residues which threaten the safety of humans. Consequently, the use of antibiotic growth promoters in feed for livestock and poultry has been banned in Japan, Korea and the European Union [3]. In China, the addition of antibiotic growth promoters to livestock and poultry feed was also forbidden in 2020 [4]. Forbidding the use of antibiotic growth promoters in feed may lead to new challenges in swine production, such as lower growth performance and higher diarrhea incidence in piglets. However, little is known about the optimal weaning age for piglets when dietary antibiotics are not available postweaning. In the European Union, 28 d is the normal age at weaning (welfare legislation) but 21 d is the minimum if specific structures for weaned piglets are available [5].

If piglets are weaned too late, the reproductive efficiency of sows is compromised [6]. Alternatively, if piglets are weaned too early, weaning stress caused by changes in feed and the environment will have a great negative impact on the growth and development of piglets [7]. Weaning induces oxidative stress in piglets regardless of weaning age [8]. Growth performance of pigs weaned at 14–23 days of age was significantly lower than that of pigs weaned at 28–35 days of age [9,10,11]. When using dietary antibiotics, growth performance of weaned pigs increases linearly with increasing weaning age to 25 d, which seems to be the optimal weaning age [12,13]. Weaning stress causes pigs to produce a high concentration of free radicals which destroys the redox equilibrium of pigs [14]. This condition leads to intestinal epithelial cell damage, and destruction of intestinal morphology and structure in weaned pigs. These changes depress feed intake and growth rate and can lead to diarrhea, inflammatory reactions, death and other phenomena in weaned pigs [15].

The weaning period is also an important period for the intestinal development of pigs. During this period, digestion, immunity, metabolism and other aspects of pigs change rapidly. The damage to intestinal morphology and structure caused by weaning is reconstructed during the maturation of intestinal function [16]. Weaning impairs the intestinal barrier of pigs. Compromised intestinal barrier function easily leads to intestinal allergy and activation of some neural regulatory pathways. At this time, viruses and bacteria can invade the intestine and cause diarrhea in piglets [17,18,19].

Therefore, selection of the optimal weaning age is crucial to the success of pig production in a future that does not permit antibiotic use in diets. The purpose of this study was to investigate the growth performance, intestinal morphology and antioxidant activity of piglets weaned at 21 or 28 d fed diets without antibiotic growth promoters.

## 2. Materials and Methods

The protocol (No. AW30301202-1-1) employed in this trial was approved by the China Agricultural University Animal Care and Use Committee (Beijing, China).

### 2.1. Pigs and Experimental Protocol

Duroc × Landrace × Large White piglets (n = 96) were random selected at 14 days old from 24 litters based on body weight and sex. All piglets were allocated to 2 groups in a completely random design with six replicate pens and eight pigs per replicate pen (four barrows and four gilts). Additionally, piglets continued to breastfeed from the original dam until weaning. Pigs were weaned at 21 (n = 48; BW = 6.87 ± 0.33 kg) or 28 (n = 48; BW = 8.49 ± 0.41 kg) days of age.

After weaning, pigs were fed a corn–soybean meal-based diet (without growth promoter) (Table 1). The basal diet was formulated to meet or exceed NRC (2012) recommendations for weaned pigs [20]. Pigs were housed in pens with a totally slatted floor (1.2 × 2.0 m) containing a nipple drinker and stainless-steel feeder for ad libitum access to water and feed over the 28 d trial. An ambient room temperature was maintained at 29 °C for the first week, and lowered by 1 °C each week thereafter. Fecal scores were recorded daily for all pigs from d 1 to 28 by the same person and were based on the following scale: 1 = well-formed feces, 2 = sloppy feces, 3 = diarrhea [21]. Pigs with a score of 3 were considered to have diarrhea. The incidence of diarrhea for piglets in each pen was calculated as [(number of piglets with diarrhea × number of days of diarrhea)/(total number of piglets × number of days of experiment)] × 100 [22]. Body weight was measured on d 0, 7, 14 and 28 and feed disappearance on these days was recorded. ADG, ADFI and F:G were calculated based on weight measurements. The final body weight was measured at 56 days of age.

### 2.2. Chemical Analysis

The ingredients and diet were analyzed according to the AOAC (2012) procedure [23], including crude protein, total phosphorus and calcium. For analysis of most amino acids, the ingredients and diet were hydrolyzed in 6 M HCl for 24 h at 110 °C. Determination of sulfur amino acid content was carried out after formic acid oxidation (AOAC, 2012). Amino acid analysis was carried out using a liquid chromatograph (Hitachi L-8800 Amino Acid Analyzer, Tokyo, Japan).

### 2.3. Sample Collection and Processing

After an overnight fast on d 14 and d 28 postweaning, a total of 24 pigs (2 pigs per pen) were selected for blood sampling. Selected pigs had body weight closest to the mean body weight of all pigs within the pen. Blood samples were collected from the jugular vein into a vacuum tube without anticoagulant for serum. Serum was separated by centrifugation for 10 min at 3000× *g* and 4 °C, and stored at −20 °C until analysis.

On d 7, d 14 and d 28 postweaning, a total of 36 pigs (12 pigs per time point; 1 pig per pen) were slaughtered using electrical stunning. Immediately after slaughter, the gastrointestinal tract was carefully removed by dissection. The pH of the stomach, duodenum, jejunum, ileum, cecum and colon contents was measured with an SPK pH meter (pH-star, DK2730, Herlev, Denmark). Tissue samples from the middle of the duodenum, jejunum and ileum were harvested (approximately 1 to 2 cm) and fixed with 4% formaldehyde–phosphate buffer and kept at 4 °C for a microscopic assessment of mucosal morphology. Heart, liver, kidney and jejunum were collected and quickly frozen in liquid nitrogen. Tissue samples were then stored at −80 °C

### 2.4. Measurement of Intestinal Morphology

Villus height and crypt depth in the duodenum, jejunum and ileum were determined. Cross-sections of intestinal samples were fixed in 4% paraformaldehyde for 24 h and then embedded in paraffin wax. Sections 4 μm thick were cut and stained with hematoxylin and eosin. In each cross section of tissue, at least 6 complete villous–crypt structures were examined under a microscope, and villous height and crypt depth were measured using an Image Pro-Plus 6.0 Software Analysis System (Media Cybernetics, Singapore).

### 2.5. Antioxidative Physiological Analyses

Activity of total superoxide dismutase (T-SOD), glutathione peroxidase (GSH-Px), catalase (CAT), total antioxidant capacity (T-AOC) and malondialdehyde (MDA) in serum, heart, liver, kidney and jejunum were determined using assay kits according to the manufacturer’s instructions (Nanjing Jiancheng Bioengineering Institute, Nanjing, China).

### 2.6. Statistical Analysis

All experimental data were analyzed using the GLM procedure in SAS (SAS Inst. Inc., Cary, NC, Canada) and repeated measurements were considered when analyzing the effects of growth performance. Mean values are reported as LS mean ± S.E. Differences in diarrhea incidence among treatments were tested by a chi-square test. Pen was the experimental unit for performance traits, and individual pig was the experimental unit for all other traits studied. Additionally, all data except diarrhea incidence of the experiment followed a normal distribution. Treatment effects were considered significant at *p* < 0.05.

## 3. Results

### 3.1. Performance and Diarrhea Incidence

The effects of weaning age on ADG, ADFI and F:G are presented in Table 2. Pigs weaned at 28 d had an increased ADG from d 8 to 14 (*p* < 0.01) than pigs weaned at 21 d. Pigs weaned at 28 d had a higher ADFI from d 0 to 7 (*p* < 0.01), d 8 to 14 (*p* < 0.01), d 15 to 28 (*p* < 0.05) and during the entire experimental period (*p* < 0.01) than pigs weaned at 21 d. Pigs weaned at 21 d had an improved F:G from d 15 to 28 (*p* < 0.05) compared with pigs weaned at 28 d. No difference was observed in final body weight (56 d) among the two groups (*p* > 0.05).

Diarrhea incidence is exhibited in Table 3. Pigs weaned at 28 d had a decreased incidence of diarrhea compared with pigs weaned at 21 d from d 8 to 14 (*p* < 0.01) and the entire experimental period (*p* < 0.01).

### 3.2. Gastrointestinal pH of Piglets

Results on pH in the gastrointestinal tract are presented in Table 4. On d 28, the pH of stomach contents from pigs weaned at 21 d was significantly higher compared with pigs weaned at 28 d (*p* < 0.01). During the entire experimental period, there was no significant difference in pH of the content of the jejunum, ileum, cecum and colon.

### 3.3. Villous Morphology in the Small Intestine

Effects of weaning age on villous morphology in the small intestine of weaned pigs are presented in Table 5, Table 6 and Table 7. In the duodenum, pigs weaned at 28 d showed greater villus height:crypt depth (*p* < 0.05) on d 14 compared with pigs weaned at 21 d. In the jejunum, pigs weaned at 28 d showed greater villus height and villus height:crypt depth (*p* < 0.05) on d 14 compared with pigs weaned at 21 d. In the ileum, pigs weaned at 28 d expressed increased villus height (*p* < 0.05) on d 14 compared with pigs weaned at 21 d.

### 3.4. Redox Status

Weaning age influenced antioxidative enzymes in many tissues. In the heart, pigs weaned at 28 d had increased activities of T-SOD (*p* < 0.05) and decreased concentration of MDA (*p* < 0.05) on d 14 and increased activities of CAT (*p* < 0.05) and decreased concentration of MDA (*p* < 0.01) on d 28 compared with pigs weaned at 21 d (Table 8). In the liver, pigs weaned at 28 d had increased activities of T-SOD (*p* < 0.05) and decreased concentration of MDA (*p* < 0.01) on d 14 and 28 compared with pigs weaned at 21 d (Table 9). In the kidney, pigs weaned at 28 d had increased activities of T-SOD (*p* < 0.05), GSH-Px (*p* < 0.05) and CAT (*p* < 0.05) and decreased concentration of MDA (*p* < 0.05) on d 14 compared with pigs weaned at 21 d (Table 10). In the jejunum, pigs weaned at 28 d had increased activities of T-SOD (*p* < 0.05) and decreased concentration of MDA (*p* < 0.05) on d 14 and increased activities of GSH-Px (*p* < 0.01) and CAT (*p* < 0.01) and decreased concentration of MDA (*p* < 0.01) on d 28 compared with pigs weaned at 21 d (Table 11). In serum, pigs weaned at 28 d had increased activities of T-SOD (*p* < 0.05) and T-AOC (*p* < 0.01) and decreased concentration of MDA (*p* < 0.05) on d 14 compared with pigs weaned at 21 d (Table 12).

## 4. Discussion

Weaning is an inevitable process for pigs. Due to the various changes in diet and the environment during weaning, pigs experience intense stress. Pigs subjected to these stresses typically reduce daily feed intake and weight gain and experience increased occurrence of diarrhea. Weaning stress is longer and more intense when piglets are weaned at an earlier age compared with a later age [24]. Daily feed intake of pigs weaned at 28 d was increased during the entire experimental period compared with pigs weaned at 21 d. From d 8 to 14, the ADG of pigs weaned at 28 d was greater compared with pigs weaned at 21 d. These findings are consistent with other studies showing that improvements in ADG and ADFI after weaning are associated with increased weaning age [9,25]. Importantly, from d 0 to 7, pigs weaned at 21 d lost weight. This result suggests that younger piglets have lower ADFI and insufficient digestive capacity than pigs weaned at older ages. Stress on pigs weaned at 21 d was more severe when fed a diet without antibiotic growth promoters [26]. During the entire experimental period, there were no significant differences in F:G between the two weaning ages. This may reflect limitations in early postweaning digestibility that is independent of weaning age [3]. Moreover, there was no statistical difference between final body weights of the two groups.

Weaning is also an important period for intestinal development of pigs. In our research, pigs weaned at 28 d showed improved morphology of the duodenum, jejunum and ileum on d 14 compared with pigs weaned at 21 d. Researchers demonstrated that intestinal villus height is positively related to growth performance of pigs postweaning [27]. During weaning, reduced feed intake of pigs resulted in a lack of dietary energy and protein. The regeneration rate of intestinal epithelial cells cannot keep pace with the rate of apoptosis. As a result, villi height became shorter, and crypts became deeper. Consequently, the ratio of villus height to crypt depth, which is used to assess nutrient absorption and digestion, declined, suggesting compromised nutrient uptake [28,29,30].

In this study, the diarrhea incidence of pigs weaned at 28 d decreased from d 8 to 14 and the entire experimental period compared with pigs weaned at 21 d. The incidence of diarrhea corresponded with altered villous morphology in the small intestine. Levels of digestive enzymes in the stomach of weaned pigs were low because the gastric fundus gland could not secrete enough hydrochloric acid to support sufficient pepsin activity [31]. As a result, digestion and utilization of protein, especially of plant proteins, were inadequate. We found that the pH of the stomach decreases over time in pigs weaned at 28 d while it increases in pigs weaned at 21 d. Insufficient gastric acid secretion also leads to accelerated emptying of the stomach which introduces the inadequately digested feed into the lower portion of the digestive tract [32]. The ability of the stomach and small intestine to absorb nutrients decreases, resulting in excess nutrients entering the hindgut. Excess nutrition in the hindgut leads to excessive reproduction of harmful microorganisms, destroying the balance of intestinal microflora and causing diarrhea in piglets.

Weaning also leads to oxidative stress in pigs [14]. During weaning stress, the organs of pigs produce large amounts of free radicals. Free radicals oxidize DNA, proteins and lipids to hydrogen peroxide, which can cause organ and cell damage if produced in excess [33]. It is important to repair and maintain the antioxidant capacity of pigs to combat oxidative stress. Therefore, we measured the antioxidant capacity of several major organs, including the heart, liver, kidneys and intestines, as well as the serum. In this study, the antioxidant capacity of pigs weaned at 28 d was slightly better than pigs weaned at 21 d in the heart, liver, kidney, jejunum and serum. The antioxidant function of pigs is related to growth performance, intestinal absorption and immune function [34,35]. The antioxidant system of pigs includes antioxidant enzymes such as SOD, GSH-Px and CAT [36]. Superoxide dismutase catalyzes the efficient dismutation of O^2−^ to H_2_O_2_ which is cleaned by GSH-Px and CAT [37]. T-AOC is a biomarker for oxidative stress and antioxidant defense and is the criterion for assessing a healthy body state [38]. The antioxidant enzyme activities can effectively protect the structure and functional integrity of cell membranes, and are the main components of important antioxidant systems in the body [39]. Pigs weaned at 28 d had better antioxidant capacity than pigs weaned at 21 d. MDA is a naturally occurring product of prostaglandin biosynthesis and lipid peroxidation and is an indicator of lipid peroxidation [40]. In particular, pigs weaned at 21 d had higher concentrations of MDA in the heart, liver, kidney, jejunum and serum, which indicated that lipid peroxidation was more serious than in pigs weaned at 28 d. The result of redox status indicated that pigs weaned at 28 d suffered less weaning stress than pigs weaned at 21 d.

## 5. Conclusions

In conclusion, our results showed that intestinal morphology, pH of the stomach and antioxidant status of pigs weaned at 28 d were more robust than in pigs weaned at 21 d without antibiotic use. Weaning at 28 days was better for the growth of pigs under the current experimental condition. These results indicated that compared with pigs weaned at 28 d, one of the vitally important strategies to improve growth performance and decrease diarrhea incidence without antibiotic use for pigs weaned at 21 d is enhancing antioxidant ability. The possible effects (positive or negative) on the sows could be investigated in further studies.

## Figures and Tables

**Table 1 animals-11-02169-t001:** Composition and analyzed nutrient composition of experimental diets (as-fed basis).

Item	
Ingredients, %	
Corn	59.34
Soybean meal (45% crude protein)	15.00
Extruded soybean (36% crude protein)	5.00
Soy protein concentrate (65% crude protein)	2.00
Fish meal (68% crude protein)	3.00
Dried whey (12% crude protein)	6.00
SDPP ^1^	2.00
Soybean oil	2.20
Sucrose	2.00
Limestone	0.90
Dicalcium phosphate	1.00
Salt	0.25
L-Lysine·HCl	0.48
L-Threonine	0.16
L-Tryptophan	0.05
L-Methionine	0.12
Vitamin and mineral premix ^2^	0.50
Nutrient levels ^3^, %	
Digestible energy, MJ/kg	15.10
Crude protein	20.21
Lysine	1.54
Methionine	0.44
Threonine	0.97
Tryptophan	0.25
Calcium	0.81
Total phosphorus	0.65

^1^ SDPP: spray-dried porcine plasma. ^2^ Provided the following vitamins and minerals per kg of diet: vitamin A, 11,000 IU as retinyl acetate; vitamin D_3_, 1500 IU as cholecalciferol; vitamin E, 44.1 IU as DL-α-tocopherol acetate; vitamin K_3_, 4 mg as menadione; vitamin B_1_, 1.4 mg; vitamin B_2_, 5.2 mg; vitamin B_5_, 20 mg; vitamin B_12_, 10 μg; niacin, 26 mg; pantothenic acid, 14 mg; folic acid, 0.8 mg; biotin, 44 μg; Fe, 100 mg from FeSO_4_; Cu, 16.5 mg from CuSO_4_·5H_2_O; Zn, 90 mg from ZnO; Mn, 35 mg from MnSO_4_; I, 0.3 mg from KI; Se, 0.3 mg from Na_2_SeO_3_. ^3^ Digestible energy is calculated value. Other nutrient levels in the table are analyzed values.

**Table 2 animals-11-02169-t002:** Effect of weaning age on growth performance of weaned pigs ^1^.

Item ^2^	Weaning Age		SEM	*p* Value
	21 Days of Age	28 Days of Age		
Initial BW (14 days of age), kg	5.14	5.12	0.06	0.791
Weaning BW, kg	6.87	8.49	0.59	0.044
Final BW (56 days of age), kg	15.78	16.90	0.53	0.141
0 to 7 days after weaning				
ADG, g/d	−4	36	17.86	0.113
ADFI, g/d	93	189	23.06	<0.01
F:G	/	/	/	/
8 to 14 days after weaning				
ADG, g/d	130	239	32.48	<0.01
ADFI, g/d	260	429	38.36	<0.01
F:G	2.28	1.90	0.30	0.400
15 to 28 days after weaning				
ADG, g/d	357	379	25.29	0.559
ADFI, g/d	595	721	43.05	0.030
F:G	1.67	1.92	0.08	0.017
0 to 28 days after weaning				
ADG, g/d	230	284	21.23	0.071
ADFI, g/d	424	561	36.66	<0.01
F:G	1.87	2.00	0.09	0.293

^1^ Values are LS means of 6 pens and pooled SEM. ^2^ ADG: average daily gain; ADFI: average daily intake; F:G: feed conversion (feed:gain).

**Table 3 animals-11-02169-t003:** Effect of weaning age on diarrhea incidence of weaned pigs ^1^.

Item	Weaning Age		SEM	*p* Value
	21 Days of Age	28 Days of Age		
Diarrhea incidence, ^2^ %				
0 to 7 days after weaning	20.86	18.66	0.75	0.318
8 to 14 days after weaning	22.56	18.28	0.88	<0.01
15 to 28 days after weaning	14.32	11.87	0.72	0.189
0 to 28 days after weaning	17.59	14.77	0.70	<0.01

^1^ Values are LS means of 6 pens and pooled SEM. ^2^ Number of pigs with diarrhea in each pen × diarrhea days/(total number of pigs × 28 d) × 100.

**Table 4 animals-11-02169-t004:** Effect of weaning age on gastrointestinal pH of weaned pigs ^1^.

Item	Weaning Age		SEM	*p* Value
	21 Days of Age	28 Days of Age		
7 days after weaning				
Stomach	3.47	3.32	0.38	0.802
Jejunum	6.01	5.80	0.15	0.354
Ileum	6.59	6.60	0.20	0.991
Cecum	6.03	5.76	0.23	0.428
Colon	6.09	6.03	0.17	0.838
14 days after weaning				
Stomach	3.51	3.15	0.40	0.557
Jejunum	5.76	5.61	0.08	0.209
Ileum	6.80	6.80	0.17	0.985
Cecum	5.95	5.73	0.14	0.278
Colon	6.18	5.76	0.12	0.166
28 days after weaning				
Stomach	4.19	2.62	0.45	<0.01
Jejunum	5.74	5.73	0.10	0.957
Ileum	7.17	7.20	0.10	0.748
Cecum	5.47	5.60	0.08	0.267
Colon	5.68	5.84	0.08	0.132

^1^ Values are LS means of 6 pens and pooled SEM.

**Table 5 animals-11-02169-t005:** Effect of weaning age on morphology of duodenum in weaned pigs ^1^.

Item	Weaning Age		SEM	*p* Value
	21 Days of Age	28 Days of Age		
7 days after weaning				
Villus height, μm	197	234	26	0.343
Crypt depth, μm	124	107	10	0.220
Villus height:crypt depth	1.69	2.22	0.29	0.221
14 days after weaning				
Villus height, μm	231	293	31	0.173
Crypt depth, μm	162	126	13	0.052
Villus height:crypt depth	1.50	2.34	0.27	0.018
28 days after weaning				
Villus height, μm	377	359	29	0.692
Crypt depth, μm	160	142	12	0.338
Villus height:crypt depth	2.36	2.55	0.14	0.354

^1^ Values are LS means of 6 pens and pooled SEM.

**Table 6 animals-11-02169-t006:** Effect of weaning age on morphology of jejunum in weaned pigs ^1^.

Item	Weaning Age		SEM	*p* Value
	21 Days of Age	28 Days of Age		
7 days after weaning				
Villus height, μm	323	352	38	0.609
Crypt depth, μm	124	115	9	0.570
Villus height:crypt depth	2.71	3.11	0.39	0.500
14 days after weaning				
Villus height, μm	241	343	36	0.045
Crypt depth, μm	126	114	6	0.252
Villus height:crypt depth	1.96	3.01	0.34	0.018
28 days after weaning				
Villus height, μm	380	384	29	0.939
Crypt depth, μm	142	113	14	0.148
Villus height:crypt depth	2.86	3.48	0.40	0.295

^1^ Values are LS means of 6 pens and pooled SEM.

**Table 7 animals-11-02169-t007:** Effect of weaning age on morphology of ileum in weaned pigs ^1^.

Item	Weaning Age		SEM	*p* Value
	21 Days of Age	28 Days of Age		
7 days after weaning				
Villus height, μm	288	291	18	0.935
Crypt depth, μm	161	132	13	0.108
Villus height:crypt depth	1.83	2.27	0.21	0.148
14 days after weaning				
Villus height, μm	290	406	38	0.024
Crypt depth, μm	120	129	15	0.686
Villus height:crypt depth	2.70	3.25	0.46	0.422
28 days after weaning				
Villus height, μm	381	422	42	0.395
Crypt depth, μm	152	156	12	0.840
Villus height:crypt depth	2.50	2.85	0.20	0.344

^1^ Values are LS means of 6 pens and pooled SEM.

**Table 8 animals-11-02169-t008:** Effect of weaning age on heart antioxidant indexes of weaned pigs ^1^.

Item ^2^	Weaning Age		SEM	*p* Value
	21 Days of Age	28 Days of Age		
14 days after weaning				
T-SOD, U/mg	107.98	126.74	5.39	<0.01
GSH-Px, U/mg	639.20	698.14	46.58	0.433
CAT, U/mg	0.96	1.03	0.07	0.547
T-AOC, U/mg	2.62	2.27	0.23	0.324
MDA, nmol/mg	2.96	1.90	0.35	0.033
28 days after weaning				
T-SOD, U/mg	112.96	116.47	3.46	0.523
GSH-Px, U/mg	708.62	712.69	36.45	0.945
CAT, U/mg	0.65	1.20	0.18	0.025
T-AOC, U/mg	2.90	3.40	0.533	0.557
MDA, nmol/mg	2.34	1.39	0.25	<0.01

^1^ Values are LS means of 6 pens and pooled SEM. ^2^ T-SOD: total superoxide dismutase; GSH-Px: glutathione peroxidase; CAT: catalase; T-AOC: total antioxidant capacity; MDA: malondialdehyde.

**Table 9 animals-11-02169-t009:** Effect of weaning age on liver antioxidant indexes of weaned pigs ^1^.

Item ^2^	Weaning Age		SEM	*p* Value
	21 Days of Age	28 Days of Age		
14 days after weaning				
T-SOD, U/mg	60.09	67.67	2.43	0.039
GSH-Px, U/mg	600.34	632.98	38.70	0.623
CAT, U/mg	2.46	2.69	0.15	0.348
T-AOC, U/mg	3.67	4.43	0.38	0.221
MDA, nmol/mg	7.15	6.11	0.27	<0.01
28 days after weaning				
T-SOD, U/mg	58.57	66.51	2.76	0.045
GSH-Px, U/mg	539.82	603.17	41.39	0.301
CAT, U/mg	2.29	2.06	0.20	0.449
T-AOC, U/mg	4.11	3.70	0.29	0.340
MDA, nmol/mg	7.08	5.81	0.36	<0.01

^1^ Values are LS means of 6 pens and pooled SEM. ^2^ T-SOD: total superoxide dismutase; GSH-Px: glutathione peroxidase; CAT: catalase; T-AOC: total antioxidant capacity; MDA: malondialdehyde.

**Table 10 animals-11-02169-t010:** Effect of weaning age on kidney antioxidant indexes of weaned pigs ^1^.

Item ^2^	Weaning Age		SEM	*p* Value
	21 Days of Age	28 Days of Age		
14 days after weaning				
T-SOD, U/mg	77.96	96.70	6.45	0.032
GSH-Px, U/mg	655.46	794.84	47.12	0.037
CAT, U/mg	3.29	4.19	0.30	0.026
T-AOC, U/mg	3.84	5.07	0.50	0.085
MDA, nmol/mg	5.29	3.21	0.72	0.033
28 days after weaning				
T-SOD, U/mg	79.00	82.84	5.49	0.643
GSH-Px, U/mg	671.62	703.04	35.13	0.553
CAT, U/mg	3.61	3.40	0.26	0.595
T-AOC, U/mg	4.83	4.27	0.24	0.186
MDA, nmol/mg	4.89	5.05	0.46	0.818

^1^ Values are LS means of 6 pens and pooled SEM. ^2^ T-SOD: total superoxide dismutase; GSH-Px: glutathione peroxidase; CAT: catalase; T-AOC: total antioxidant capacity; MDA: malondialdehyde.

**Table 11 animals-11-02169-t011:** Effect of weaning age on jejunum antioxidant indexes of weaned pigs ^1^.

Item ^2^	Weaning Age		SEM	*p* Value
	21 Days of Age	28 Days of Age		
14 days after weaning				
T-SOD, U/mg	60.19	80.08	7.23	0.045
GSH-Px, U/mg	544.63	484.12	41.52	0.326
CAT, U/mg	2.62	2.72	0.18	0.698
T-AOC, U/mg	3.78	3.60	0.39	0.763
MDA, nmol/mg	6.19	3.65	0.68	<0.01
28 days after weaning				
T-SOD, U/mg	60.53	66.08	6.18	0.551
GSH-Px, U/mg	369.81	667.38	63.09	<0.01
CAT, U/mg	2.50	3.25	0.22	<0.01
T-AOC, U/mg	3.14	3.42	0.35	0.613
MDA, nmol/mg	6.50	4.14	0.71	<0.01

^1^ Values are LS means of 6 pens and pooled SEM. ^2^ T-SOD: total superoxide dismutase; GSH-Px: glutathione peroxidase; CAT: catalase; T-AOC: total antioxidant capacity; MDA: malondialdehyde.

**Table 12 animals-11-02169-t012:** Effect of weaning age on serum antioxidant indexes of weaned pigs ^1^.

Item ^2^	Weaning Age		SEM	*p* Value
	21 Days of Age	28 Days of Age		
14 days after weaning				
T-SOD, U/mL	147.94	153.43	1.77	0.019
GSH-Px, U/mL	826.14	829.41	35.80	0.952
CAT, U/mL	5.96	6.06	0.55	0.906
T-AOC, U/mL	6.31	8.74	0.68	<0.01
MDA, nmol/mL	4.32	3.79	0.20	0.050
28 days after weaning				
T-SOD, U/mL	153.37	151.33	2.36	0.566
GSH-Px, U/mL	787.58	881.70	42.65	0.123
CAT, U/mL	5.91	6.76	0.49	0.240
T-AOC, U/mL	7.34	6.37	0.44	0.174
MDA, nmol/mL	4.01	3.84	0.30	0.706

^1^ Values are LS means of 6 pens and pooled SEM. ^2^ T-SOD: total superoxide dismutase; GSH-Px: glutathione peroxidase; CAT: catalase; T-AOC: total antioxidant capacity; MDA: malondialdehyde.

## Data Availability

Some or all data, models or code generated or used during the study are available in a repository or online in accordance with funder data retention policies (Provide full citations that include URLs or DOIs.)

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
