# Peer review of "Effects of Weaning Age at 21 and 28 Days on Growth Performance, Intestinal Morphology and Redox Status in Piglets"

_animals, 2021, doi:10.3390/ani11082169_

Round 1

Reviewer 1 Report

I have no further comments. 

Author Response

We thank the careful reading and thoughtful comments on our draft. The comments of you are highly insightful and facilitate us to improve the quality of our manuscript.

Reviewer 2 Report

With the aim that the authors set, it seems to me that the study has little meaning: it is a comparison between two weaning ages and all the other constant factors: it is clear that the result is positive for piglets weaned at 28 d (also because fed with a diet formulated for 5-7 kg LW). On the other hand, in my opinion, it could be interesting if set up to evaluate the effects of the different ages at weaning that occur in practice, where farrowing induction is not applied. In this case the Authors should however explain why they opted for a 5-7 kg LW diet and not a 7-11 kg LW diet, which would then be the most likely to occur.

table 1.

  • extruded soybeans, , I think, and not 'Extruded soybean meal'
  • standardized ileal digestible amino acids  

Table 3. how do the authors explain a pH 4.19 to 28 d for group 21 ?

The conclusions are quite obvious but even here I believe it depends on the goal that the authors have set themselves

Author Response

(The authors gave the same response as above.)

Round 2

Reviewer 2 Report

According to the Authors the purpose of the study was to investigate the growth performance, intestinal morphology and antioxidant activity of 80piglets weane dat 21 or 28 d fed diets without antibiotic growth promoters.

The experimental protocol adopted cannot provide an answer: it is not possible to conclude that 28d is better than 21d when the diet used is formulated according to the NRC requirements for piglets of 7-11 kg LW (28d weaned piglets).

Author Response

We thank the careful reading and thoughtful comments on our draft. 

This manuscript is a resubmission of an earlier submission. The following is a list of the peer review reports and author responses from that submission.

Round 1

Reviewer 1 Report

Review of Animals paper on weaning at 21 vs 28 days by Ming et al.

General comments

The topic of research is relevant and provides knowledge with both scientific and practical implications. A major concern is that the statistical model did not include litter and that at several occasions the authors drew conclusions not supported by significant P-values of the data. Please go through the results and discussion and see if statements are supported by P-values. Please address a higher ADFI as higher or greater, not improved. Please address a low F:G is an improved F:G.

Specific comments

Title: ‘…  status in pigs’ or ‘… status in piglets’

L12: delete first sentence

L16-17: delete sentence

L19: ‘These results indicated…’

L24: delete kg and insert BW at d 21 and 28 in L27 instead.

L28 and throughout: F:G not F: G

L31: ‘.. had a higher ADFI…

L33: ‘… had in improved F:G

L36: ‘… was significantly higher compared…’

L41-43: These factors supported better growth…

L85-87: rewrite

L89: delete aged

L105: ‘… based on weight measurements.’

L113-115: rewrite. unclear.

L130: spell out

L131: ‘… using assay kits according to…’

L135: analyzing

L136: not clear why sow was not included as random effect in a mixed model. There may very well be a carry over effect of sow/litter.

L145: an higher ADFI

L147: improved G:F

L150: is exhibited in

L154: ‘Results on pH in the gastrointestinal tract are presented in Table 4’

L157: ‘.. in pH of the content of jejunum…’

L160: rewrite

L163 delete And

Table 1: Extruded soybean meal? Consider to include more info on the ingredients (soy protein concentrate, fish meal, whey can vary). 3.61 MJ DE/kg seems low, please check.

All tables: Please delete ‘of age’. Please delete ANOVA. Are results least squares means instead of means? n=6 is redundant.

LL296: delete And

L302-303: include reference

307: When there is no difference you cannot also write that some were 1.12 kg heavier.

L308-309: it is the ADG not BW at 56 days, and it was a tendency.

L311-312: this is beyond what the data supports as only few items were significantly different.

L320: d 8 to 14, not d 0 to 14.

L322-329: unclear if all this relates to ref no 29

L327-328: rewrite

L332: I think you need to rewrite this because an ADFI of 93 g is not ‘a lot of energy’.  I suggest to address the low ADFI as a major challenge after weaning because higher FI is becoming clear to be a major issue in preventing diarrea.

L337-346: Please describe the analyzed items so the reader can interpret if a high or low expression is god or bad. In the current text only MDA is explained.

L347-348: delete

Author Response

Thanks for your comments. We have carefully considered your comments and revised the manuscript.

Reviewer 2 Report

The aim of this study was to investigate the growth performance, intestinal morphology and antioxidant activity of piglets weaned at 21 or 28 d fed. 

The object of the study is certainly not new or innovative but the application in the current conditions without the use of antibiotics can provide a useful contribution.

The use of the same feed for 21 and 28d weaned piglets is probably the main limitation of the study. the formulation adopted is probably too rich for piglets weaned at 28d but somewhat limiting for those at 21d.

Secondly, all the statistical comparisons made are at the same time from weaning but with pigs of different ages. Correct? I believe some caution is needed in concluding

Motivation and meaning of the study of antioxidant parameters in serum, heart, liver, kidney, and jejunum must be reported

Major

Material & methods: there is a complete lack of references to the litter size which, as is well known, affects the milk production of the sow and consequently the live weight of the piglets at weaning. At the same age there can be significant differences in LW and therefore different weaning stresses

Statistic analysis. the Authors have chosen to make a comparison of the treatments at the different post-weaning times, I believe that a time x treatment evaluation would provide much more information in order to highlight the post weaning time effect. An obvious example of the contribution that a similar analysis would provide comes for example from the evaluation of gastric pH: 2 opposite increasing trends in the 2 treatments (however quite clear)

Conclusion
The conclusions reported could perhaps have been made even without conducting the study: the Authors should try to bring the concrete conclusions reached.
No reference is ever made to possible effects (positive or negative) on the sow, on potential improvements in the energy balance, etc.

Minor

Line 56. ‘increased weaning stress’;: specify

Line 57-58. In EU 28 d is the normal age at weaning (welfare legislation) but 21 d is minimum if specific structures for weaned piglets are availables

Line 59. ‘the reproductive efficiency of sows is compromised.’ Insert references

Table 1

‘Other nutrient levels in the table were analyzed values’: microelements and amino acids ? must be reported in M&M sampling and analysis mode

Table 2

It is not clear to me: for piglets weaned at 21 days + 28 days of the experimental period, they reach an age of 49 days and not 56

Table 4. An analysis of the reported values shows an opposite trend in the variation of pH over time. As physiologically reliable, the pH decreases over time in the 28d group while it increases in the 21d group. Authors should try to explain

Author Response

(The authors gave the same response as above.)

Round 2

Reviewer 1 Report

Review of revised version (R1):

The authors replied that presented data are means and not least squares means. I believe it is very unusual to present simple means of data being statistically evaluated. I think authors need to update tables with least squares means and subsequently the results and discussion sections with the correct data.

Reviewer 2 Report

many unsatisfactory aspects

Response 2: We appreciate this comment. The basal diet was formulated to exceed NRC (2012) recommendations for weaned pigs (5-7kg). And we will consider using different feed for 21 and 28d weaned piglets in future studies.

The basal diet was NOT formulated to exceed NRC (2012) recommendations for weaned pigs (5-7kg). Methionine and threonine are also lower than NRC requirements !

The first limiting point remains unanswered.

Add in table 1 SDI if the amino acids are expressed as standardized ileal digestible.

Response 4: It has been corrected as suggested in Line 334-346.

no reason is given.

2 unanswered question

Point 6: Statistic analysis. the Authors have chosen to make a comparison of the treatments at the different post-weaning times, I believe that a time x treatment evaluation would provide much more information in order to highlight the post weaning time effect. An obvious example of the contribution that a similar analysis would provide comes for example from the evaluation of gastric pH: 2 opposite increasing trends in the 2 treatments (however quite clear)

Response 6: It has been corrected as suggested in Line 326-329.

3 unanswered question: statistic analysis.

No hypothesis is provided on the abnormal pH trend in piglets weaned at 21d. The value measured at 28d would seem to derive from some kind of error:

Point 7:

the reasons given by the Authors in answer 7 in my opinion could have been the true, correct, reason for the work, that is, the fact that in swine production, where there is no farrowing synchronization, the piglets at weaning have different ages. With all the consequences of the case, including some of those studied by the Authors.